# HOI-PAGE: Zero-Shot Human-Object Interaction Generation with Part Affordance Guidance

## Abstract

We present HOI-PAGE, a new approach to synthesizing 4D human-object interactions (HOIs) from text prompts in a zero-shot fashion, driven by part-level affordance reasoning. In contrast to prior works that focus on global, whole body-object motion for 4D HOI synthesis, we observe that generating realistic and diverse HOIs requires a finer-grained understanding – at the level of how human body parts engage with object parts. We thus introduce Part Affordance Graphs (PAGs), a structured HOI representation distilled from large language models (LLMs) that encodes fine-grained part information along with contact relations. We then use these PAGs to guide a three-stage synthesis: first, decomposing input 3D objects into geometric parts; then, generating reference HOI videos from text prompts, from which we extract part-based motion constraints; finally, optimizing for 4D HOI motion sequences that not only mimic the reference dynamics but also satisfy part-level contact constraints. Extensive experiments show that our approach is flexible and capable of generating complex multi-object or multi-person interaction sequences, with significantly improved realism and text alignment for zero-shot 4D HOI generation.

## 1 Introduction

> "The *affordances* of the environment are what it *offers* the animal, what it *provides* or *furnishes*, either for good or ill. ... It implies the complementarity of the animal and the environment." – James J. Gibson

Human-object interaction (HOI) is a fundamental aspect of everyday life, ranging from simple activities like picking up a cup to complex activities like ironing a shirt. These interactions are essential to understanding and synthesizing realistic scenes, reflecting the complex nature of object affordances (Gibson, 2014). Modeling the interaction dynamics between humans and objects is crucial for many downstream applications in computer vision and graphics, such as character animation, immersive VR/AR, robotics, and product design. In this work, we focus on generating diverse and realistic HOI motions, beyond a limited taxonomy of interactions, from easy-to-use text prompts.

Generating realistic interactions remains challenging for machines, as it requires joint understanding of object affordances, human body movements, and resulting object motions. Even a seemingly simple action, like carrying a briefcase, requires understanding that the briefcase handle affords hand grasping, the human arm swings, and the briefcase follows the hand's trajectory. Prior works typically model interactions as overall full-body and object motions, making such complex interaction modeling challenging (Li et al., 2024a; Diller & Dai, 2024; Kim et al., 2025). In contrast, our key insight is that finer-grained modeling of *part-level affordances* – how specific object parts relate to human body parts – play a crucial role in generating more realistic, diverse interactions. Moreover, this leads to a general formulation for human-object interactions that extends beyond the single-person, single-object scenarios tackled by the state-of-the art (Peng et al., 2025; Li et al., 2024b), and enables synthesis of multi-person, or multi-object HOIs (Fig. 1).

Collecting annotations for such part affordance, in order to train a supervised model, is expensive and time-consuming, due to the vast variety of 3D objects and the diversity of human interactions. While existing methods (Peng et al., 2025; Li et al., 2024b; Diller & Dai, 2024) have made significant progress in HOI generation, they rely heavily on captured 4D interaction data for supervision.

"A person lifting up a barbell from the ground with both hands"

"A person singing rock and roll while holding onto a microphone stand"

"A person skateboarding and sliding forward slowly"

"Two persons carrying a portable flat stretcher together, one in the front and the other in the back, holding the handles while moving forward"

"A person moving an iron on an ironing board while standing"

Single-Person Single-Object Interaction Generation

Multi-person Single-Object Interaction Generation

Single-Person Multi-object Interaction Generation

Figure 1: We propose to model complex 4D human-object interactions (HOIs), by inferring part affordance graphs (PAGs) that guide zero-shot HOI synthesis from a text prompt and 3D object model(s). Our PAGs, distilled from large language model reasoning, provide localized affordance constraints for our optimization-based generation, enabling flexible modeling of diverse interaction scenarios involving multiple people or objects in a zero-shot fashion.

We present HOI-PAGE, a zero-shot approach to generating realistic 4D HOI motion sequences from text prompts, covering diverse interaction scenarios involving multiple people or objects. Key to our approach is the distillation of part-level affordance graphs (PAGs) from a large language model (LLM) (Guo et al., 2025) to guide the interaction generation process through three stages: 3D object part segmentation, HOI video synthesis, and 4D HOI fitting optimization.

Given as input a set of 3D objects and a text prompt describing the desired interaction, our approach generates both human and object motion sequences performing the interaction. We first reason about affordances between object parts and human body parts by leveraging an LLM to imagine plausible human and object motions, along with their physical contact, based on the input text prompt. The resulting part-level affordances are represented as a graph, where nodes correspond to object and human body parts, and edges encode contact relationships. The inferred PAG then guides three generation stages: (1) Decomposing 3D object geometry into geometric parts; (2) Generating an HOI reference video from the text prompt and estimating object masks, depths, and 4D human motions from the video; (3) Formulating a part affordance-guided optimization to infer 4D object motions from the video, while enforcing part-level contact constraints.

Our part affordance-guided approach is flexible and generalizes well to complex interaction scenarios by easily expanding the PAG to include part nodes for multiple people or objects. We demonstrate the effectiveness of our approach through extensive experiments on a variety of interaction scenarios, including single and multi-person/object interactions (Fig. 1). Perceptual studies show that our method significantly outperforms state-of-the-art methods (Peng et al., 2025; Li et al., 2024b) in terms of interaction realism and alignment with text prompts.

**Contributions.** (1) We present the first approach that explicitly models part-level affordance guidance to enable realistic, zero-shot 4D HOI synthesis. Our method distills structured part affordance graphs from an LLM, capturing how humans interact with specific object parts. These inferred graphs guide the synthesis across multiple stages—3D object part segmentation, HOI reference video generation, and 4D HOI optimization—resulting in diverse and physically plausible interactions. (2) We formulate a part affordance-guided optimization that lifts HOI motions in reference videos to 4D, leading to accurate part-level contact in the synthesized human and object motion sequences. (3) Our part affordance graphs are flexible and versatile, enabling generalization to diverse interaction scenarios, including multi-person/object interactions. We will release our code and data.

## 2 RELATED WORK

**Human Motion Generation.** 4D human motion synthesis has seen significant advances in recent years, largely driven by advances in deep learning. Earlier work leveraged recurrent neural networks for synthesis (Fragkiadaki et al., 2015; Aksan et al., 2019; Gopalakrishnan et al., 2019; Martinez et al., 2017). More recently, with the success of denoising diffusion models (Sohl-Dickstein et al., 2015; Song et al., 2021; Ho et al., 2020), diffusion-based human motion generation has become a powerful and widely-adopted approach to synthesizing human motion (Zhang et al., 2023b; Raab et al., 2023; Zhao et al., 2023; Dabral et al., 2023; Tevet et al., 2023; Shafir et al., 2023; Zhang et al., 2022a; Karunratanakul et al., 2024; Jiang et al., 2023a; Petrovich et al., 2024). These methods show remarkable motion synthesis results, but focus on modeling human motion in isolation, without interactions intrinsic to everyday, real-world scenarios.

**Human-Object Interaction Generation.** As interactions play a crucial role in 4D synthesis, various approaches have focused on modeling human-object interactions, generating the motion of a single human and single object. Several works tackled this task under the assumption of a static object (Taheri et al., 2022; Tendulkar et al., 2023; Zhang et al., 2022b; Wu et al., 2022; Lee & Joo, 2023; Zhang et al., 2023a; Kulkarni et al., 2023), focusing only on human motion generation. Recently, new methods have proposed to generate both human and object motion for single-human single-object scenarios (Li et al., 2023; Wan et al., 2022; Diller & Dai, 2024; Peng et al., 2025; Li et al., 2024b; Wu et al., 2024; Xu et al., 2024; 2023; Wang et al., 2023; Yang et al., 2024a). These methods can synthesize realistic human-object interactions, but rely on ground truth real-world captures of human-object interaction data to train the generative models. Collecting such 4D ground truth data is very time-consuming and expensive, and thus very limited in size and diversity (typically single digit thousands of sequences with limited diversity of objects (Bhatnagar et al., 2022; Taheri et al., 2020; Jiang et al., 2023b)). In contrast, our approach proposes a general approach to handle various novel, diverse objects without requiring any 4D interaction data for training.

GenZI (Li & Dai, 2024) recently introduced a new paradigm for 3D human-scene interaction synthesis, by distilling priors from text-to-image foundation models to generate interactions without requiring 3D interaction training data, focusing only on static interaction generation (Zhang et al., 2025; Zhu et al., 2024; Yang et al., 2024b; Kim et al., 2024). Concurrent to our approach, ZeroHSI (Li et al., 2024a), DAViD (Kim et al., 2025), and ZeroHOI (Lou et al., 2025) have begun to address the challenge of zero-shot 4D human-object interaction synthesis to circumvent the need for 4D ground truth training data. While these approaches also leverage knowledge from large video foundation models, they treat the human-object motion globally, lacking finer-grained interaction modeling at the level of parts. This limits the ability to capture complex contact dynamics and multi-object or multi-person interactions. In contrast, our approach proposes to explicitly model part affordances to guide synthesis, enabling generation of complex multi-interaction scenarios.

**3D Affordance Analysis.** Various works have also proposed to study 3D affordances via structured graph representations to capture relations between humans and objects. PiGraphs (Savva et al., 2016) introduced a prototypical interaction graph representation to capture physical contact and visual attention relations between human body parts and 3D scenes, in order to synthesize static snapshots of human-scene interactions. In contrast to the graph-based representation, Fisher et al. (Fisher et al., 2015) propose an activity heatmap representation learned from human-scene interactions for synthesizing new 3D scenes that enable similar interactions. iMapper (Monszpart et al., 2019) instead proposes to leverage "scenelets" that capture short interaction subsequences as a database prior to reconstruct a human and the objects interacted with from monocular video observations. Inspired by these methods, we also propose to explicitly model affordance relations, as part-based affordance graphs of (multi-) human-object interactions for zero-shot 4D human-object interaction synthesis.

## 3 METHOD

We aim to generate 4D sequences of humans realistically interacting with diverse 3D objects from text descriptions in a zero-shot manner. Our approach, HOI-PAGE, proposes to employ part affordance graphs (PAGs) inferred from an LLM as guidance to optimize for motion sequences of both humans and objects. The flexibility of PAGs enables us to synthesize diverse, complex HOI scenarios, including (1) single-person single-object, (2) multi-person single-object, and (3) single-person multi-object interactions (Fig. 1). Our approach is illustrated in Fig. 2.

Given as input a set of 3D objects $\{\mathcal{O}\}$ to be interacted with and a short text prompt $\Gamma$ describing the desired human interaction, HOI-PAGE generates a motion sequence $\{(\mathbf{R}_t, \mathbf{t}_t)\}_{t=1}^T$ for each object $\mathcal{O}$ and a sequence of body parameters $\{\Theta_t\}_{t=1}^T$ for each human $\mathcal{H}$, where $T$ is the number of frames in the generated interaction. Object $\mathcal{O}$ is represented by a textured 3D mesh, and human $\mathcal{H}$ is parameterized by the SMPL-X model (Pavlakos et al., 2019). To simplify the notation, we omit the indexing of objects and humans here. At time step $t$, each object pose is represented by a 3D rotation $\mathbf{R}_t$ and a 3D translation $\mathbf{t}_t$, while $\Theta_t$ represents a set of SMPL-X parameters that include body joint rotations, body shape coefficients, a global rotation, and a global translation.

We first construct a PAG, denoted as $\mathcal{G}$, using an LLM based on the input text prompt $\Gamma$ (Sec. 3.1). In this graph, nodes represent object parts and human body parts, while edges represent contact relations between them. We use $\mathcal{G}$ to inform three stages of the interaction generation process: (1)

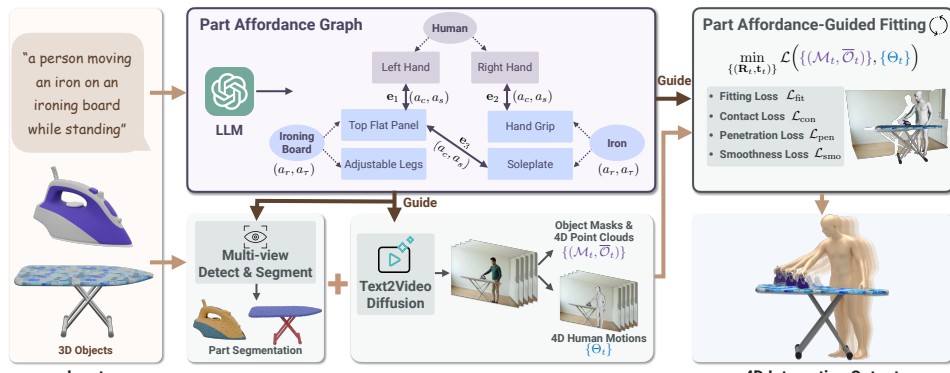

Figure 2: Our HOI-PAGE generates realistic 4D human-object interaction (HOI) motions from a given set of 3D objects and a text prompt. We introduce Part Affordance Graphs (PAGs) to capture how specific object parts relate to human body parts (**top-middle**). The PAG is distilled from a large language model (LLM) based on the text prompt and is used to guide a three-stage synthesis: (1) Decomposing the input objects into geometric parts based on multi-view detection and segmentation (**bottom-left**); (2) Generating an HOI video from the text prompt and estimating object masks, depths, and 4D human motions (**bottom-middle**); (3) Optimizing for objects motions by fitting to the video while enforcing part-level contact constraints from the PAG (**top-right**).

We apply multi-view object part segmentation to object $\mathcal{O}$ according to the corresponding part nodes in $\mathcal{G}$ (Sec. 3.2). (2) We generate an HOI reference video based on text prompt $\Gamma$ enhanced with part affordance descriptions from $\mathcal{G}$, and then estimate object masks, depths, and 4D human motions in the video (Sec. 3.3). (3) Finally, we optimize for object motions that fit to the reference video while respecting the part contact constraints from $\mathcal{G}$ (Sec. 3.4).

## 3.1 CONSTRUCTING PART AFFORDANCE GRAPHS

We define a PAG as $\mathcal{G} = (\mathcal{V}, \mathcal{E})$ consisting of node and edge sets $\mathcal{V}$ and $\mathcal{E}$, respectively. A node $\mathbf{v} \in \mathcal{V} = \mathcal{V}_o \cup \mathcal{V}_h$ can represent either object parts $\mathcal{V}_o$ or human body parts $\mathcal{V}_h$. We also add a virtual parent node $\bar{\mathbf{v}}$ to $\mathcal{V}$ to represent a whole object or human, which is connected to all its constituent part nodes. A virtual object node $\bar{\mathbf{v}}_o$ has two motion state attributes $(a_r, a_\tau)$, where $a_r$ denotes whether the object undergoes global rotation, and $a_\tau$ is analogous for undergoing global translation. If both indicators are false, the object should remain stationary throughout the interaction.

An edge $\mathbf{e} \in \mathcal{E}$ represents a contact relationship between object part nodes to a human body part node or another object part node. Each edge $\mathbf{e}$ has two attributes $(a_c, a_s)$: $a_c$ indicates whether the contact is continuous or not across the $T$ frames, while $a_s$ denotes whether the contact is relatively static or not. For example, in Fig. 2, $\mathbf{e}_2$ represents the contact between the right hand and the iron's hand grip as continuous ($a_c$ = true) and relatively static ($a_s$ = true). Edge $\mathbf{e}_3$ between the ironing board's top flat panel and the iron's soleplate is described as continuous but not static. PAGs are flexible and can represent different types of interactions, such as multi-person/multi-object interactions, by simply expanding the node sets to include other human and object parts and the edge sets to include the corresponding part contact relations.

To construct a PAG from text prompt $\Gamma$, we leverage an LLM (Guo et al., 2025) to describe the interaction motions and infer part affordances. Specifically, the LLM should infer the object part nodes $\mathcal{V}_o$, the number of humans, as well as the graph edges $\mathcal{E}$. The LLM is instructed to use a pre-defined set of 12 human body parts, including left/right hand, left/right foot, hips, among others. The LLM reasons about part segmentation labels for each object and part-level physical contact relations, and then produces all the graph nodes, edge connections, and their associated attributes.

We use an LLM for its powerful reasoning and in-context learning capabilities. We also considered using vision-language models (VLMs) for PAG inference by prompting them with interaction prompts and rendered images of 3D objects. However, we found that the VLMs we experimented with occasionally ignore the visual input in our task, partly due to the known hallucination issue (Liu et al., 2024), and they are less robust in generating plausible PAGs. We thus opt for LLMs in this work but stress that our PAG representation is agnostic to the foundation model used, and VLMs could alternatively be used as they continue to improve.

## 3.2 MULTI-VIEW OBJECT PART SEGMENTATION

Given the inferred set of part nodes $\mathcal{V}_o$ in the PAG $\mathcal{G}$, we then segment the geometry of each object $\mathcal{O}$ into the corresponding 3D semantic part segmentations, as shown in Fig. 2-bottom left. We first render $\mathcal{O}$ into images from 8 virtual camera views sampled on the viewing sphere. We then perform open-vocabulary detection using Qwen-VL (Bai et al., 2025) on the rendered images, and obtain each object part's bounding box. These boxes are used to prompt SAM2 (Ravi et al., 2024) to estimate 2D part segmentation masks for each view. We then aggregate these masks into 3D point cloud labels through voting. To simplify notation, we denote the segmented object point cloud as $\mathcal{O} = \{\mathcal{P}^o\}$, where $\mathcal{P}^o$ is the part point cloud corresponding to its part node in $\mathcal{V}_o$.

## 3.3 GENERATING HOI VIDEOS

**Generation.** We generate an HOI video $\{I_t\}_{t=1}^T$ depicting the desired interaction motions based on text prompt $\Gamma$ using an off-the-shelf video diffusion model CogVideoX (Yang et al., 2024c). To generate a video capturing the part affordances between humans and objects, we enhance $\Gamma$ to a longer prompt $\Gamma^+$ that incorporates more detailed descriptions of the contact relations from the PAG $\mathcal{G}$. We use the same LLM (Guo et al., 2025) as in Sec. 3.1 for this prompt enhancement. To further improve video generation stability and quality, we generate the first frame $I_1$ using the text-to-image model FLUX (Labs, 2024), and then leverage text+image-to-video diffusion (Yang et al., 2024c) to generate the video frames $\{I_t\}_{t=1}^T$.

In order to use the generated video frames as guidance for our 4D HOI, we extract a rich set of constraints from them. These constraints capture part-level 2D-3D object correspondence, video object geometry, as well as human poses. This is informed by 2D part segmentations, depth estimation, and human motion recovery from the video, as described below.

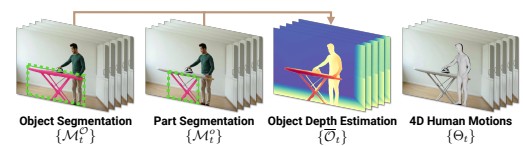

Figure 3: Inferred object constraints and human motions from a generated video.

**Video Object Part Segmentation.** We use open-vocabulary detection (Bai et al., 2025) on the first video frame to obtain bounding boxes for each object. We then track and segment objects across video frames using SAM2 (Ravi et al., 2024), using the detected bounding boxes as prompts. This produces a sequence of 2D object masks $\{\mathcal{M}_t^{\mathcal{O}}\}_{t=1}^T$ in the video for object $\mathcal{O}$, as shown in Fig. 3-left. Similarly, for each object part $\mathcal{P}^o \in \mathcal{O}$, we also compute its 2D segmentation masks $\{\mathcal{M}_t^o\}_{t=1}^T$.

**Video Depth Estimation.** We estimate depth $\{D_t\}_{t=1}^T$ for the video frames using the pre-trained depth estimation model MoGe (Wang et al., 2024). Combining $\{D_t\}$ with the per-frame object masks $\{\mathcal{M}_t\}$ and object part masks $\{\mathcal{M}_t^o\}$, we obtain a sequence of 3D point clouds for each object in the generated video: for frame $t$, we have $\overline{\mathcal{O}}_t = \{\overline{\mathcal{P}}_t^o\}$, where $\overline{\mathcal{O}}_t$ denotes the estimated object point cloud, and $\overline{\mathcal{P}}_t^o$ denotes the estimated point cloud for object part $\mathcal{P}^o$.

**4D Human Pose Estimation.** We use a state-of-the-art human motion recovery method GVHMR (Shen et al., 2024) to estimate the body parameters $\{\Theta_t\}_{t=1}^T$ for each human in the generated video. This model produces 4D human motions represented by the SMPL-X (Pavlakos et al., 2019) body parameters over time; however, it estimates the human motion only in isolation, and we need to characterize the holistic human-object interactions by considering part-level affordance constraints from the PAG $\mathcal{G}$ in our 4D HOI optimization in Sec. 3.4.

## 3.4 PART AFFORDANCE-GUIDED 4D HOI OPTIMIZATION

We formulate a part affordance-guided optimization that estimates object motion sequences $\{(\mathbf{R}_t, \mathbf{t}_t)\}_{t=1}^T$ based on the PAG $\mathcal{G}$ (Sec. 3.1), 3D object part segmentation $\{\mathcal{P}^o\}$ (Sec. 3.2), and the generated HOI video $\{I_t\}_{t=1}^T$ constraints (Sec. 3.3). The optimization is fundamentally based on our part-based affordance graph representation in order to ensure plausible motions and relations between objects and previously estimated human bodies $\{\Theta_t\}_{t=1}^T$. It aims to ensure that objects fit well to the generated video at the part level, object motions respect the part contact relations in $\mathcal{G}$ while avoiding penetration, and the resulting object motions are temporally smooth.

**Fitting Loss.** We fit 3D object $\mathcal{O}$ to each frame of the generated video $\{I_t\}_{t=1}^T$ at both object and part levels, and in both 2D and 3D. The part-level correspondence provides higher-level guidance to help to avoid poor local minima in the optimization, while low-level point correspondences help to attain finer-grained alignment. In 3D space, we compute the fitting loss for objects as:

$$\mathcal{L}_{3D}^{\mathcal{O}} = \sum_{\mathcal{O}} \sum_{t=1}^T \text{CD}(\mathbf{R}_t \mathcal{O} + \mathbf{t}_t, \overline{\mathcal{O}}_t), \tag{1}$$

where $\text{CD}(\cdot)$ denotes the Chamfer Distance (CD) between two 3D point clouds. We compute the fitting loss for each object part $\mathcal{P}^o$ as:

$$\mathcal{L}_{3D}^{o} = \sum_{\mathcal{O}} \sum_{\mathcal{P}^o} \sum_{t=1}^T \text{CD}(\mathbf{R}_t \mathcal{P}^o + \mathbf{t}_t, \overline{\mathcal{P}}_t^o). \tag{2}$$

The object-level fitting loss helps to mitigate any effect from potentially inaccurate part segmentations, while the part-level fitting loss can help to find better correspondences between the object and the generated video. Similarly, we compute the 2D fitting losses $\mathcal{L}_{2D}^{\mathcal{O}}$ and $\mathcal{L}_{2D}^{o}$ for the object and its parts, respectively. We project the 3D object point clouds $\mathcal{O} = \{\mathcal{P}^o\}$ to the image space using the estimated camera intrinsics of the generated video and compute CD losses between the projected object point clouds and the 2D object mask pixels $\{\mathcal{M}_t\}_{t=1}^T$ and 2D object part mask pixels $\{\mathcal{M}_t^o\}_{t=1}^T$, respectively. Overall, the fitting loss is $\mathcal{L}_{\text{fit}} = \mathcal{L}_{3D}^{\mathcal{O}} + \mathcal{L}_{3D}^{o} + \mathcal{L}_{2D}^{\mathcal{O}} + \mathcal{L}_{2D}^{o}$.

**Part-based Contact Loss.** We compute the contact loss on a part basis guided by each edge $\mathbf{e} = (\mathbf{v}_1, \mathbf{v}_2)$ and its attribute $a_c$ in our PAG $\mathcal{G}$:

$$\mathcal{L}_{\text{cc}} = \sum_{\mathbf{e}=(\mathbf{v}_1,\mathbf{v}_2)\in\mathcal{E}} \begin{cases} \frac{1}{T}\sum_{t=1}^T \text{MD}(\mathcal{P}_t^{\mathbf{v}_1}, \mathcal{P}_t^{\mathbf{v}_2}), & \text{if } a_c = \text{true} \\ \min_t \text{MD}(\mathcal{P}_t^{\mathbf{v}_1}, \mathcal{P}_t^{\mathbf{v}_2}), & \text{otherwise} \end{cases} \tag{3}$$

where $\mathcal{P}_t^{\mathbf{v}_1}$ and $\mathcal{P}_t^{\mathbf{v}_2}$ are the 3D part point clouds of $\mathbf{v}_1, \mathbf{v}_2$ at time step $t$ and can be either an object part or a human body part. $\text{MD}(\cdot)$ denotes the minimum distance between any pair of nearest neighbors between the two point clouds. The top case is for continuous contact across the $T$ frames, while the bottom case is for non-continuous contact. We also measure relative contact dynamics (static vs. dynamic) between $\mathbf{v}_1, \mathbf{v}_2$ based on the attribute $a_s$ of each edge $\mathbf{e}$ in the PAG $\mathcal{G}$:

$$\mathcal{L}_{\text{cd}} = \sum_{\mathbf{e}=(\mathbf{v}_1,\mathbf{v}_2)\in\mathcal{E}} \sum_t \begin{cases} \mathcal{L}_2(\mathcal{P}_t^{\mathbf{v}_2\to\mathbf{v}_1}, \mathcal{P}_{t+1}^{\mathbf{v}_2\to\mathbf{v}_1}), & \text{if } a_s = \text{true} \\ \mathcal{L}_2(\mathcal{P}_t^{\mathbf{v}_2\to\mathbf{v}_1}, \frac{1}{2}(\mathcal{P}_{t-1}^{\mathbf{v}_2\to\mathbf{v}_1} + \mathcal{P}_{t+1}^{\mathbf{v}_2\to\mathbf{v}_1})), & \text{otherwise} \end{cases} \tag{4}$$

where $\mathcal{P}_t^{\mathbf{v}_2\to\mathbf{v}_1}$ denotes the 3D part point cloud of node $\mathbf{v}_2$ at time step $t$ transformed to the canonical object space of node $\mathbf{v}_1$ by the inverse of the corresponding object pose $(\mathbf{R}_t, \mathbf{t}_t)$ of $\mathbf{v}_1$, assuming $\mathbf{v}_1$ is always an object part node. $\mathcal{L}_2(\cdot)$ measures the average Euclidean distance of each corresponding point pairs in the two point clouds. The top case promotes static contact, while the bottom case promotes dynamic but temporally coherent contact. Overall, the contact loss is $\mathcal{L}_{\text{con}} = \mathcal{L}_{\text{cc}} + \mathcal{L}_{\text{cd}}$.

**Penetration Loss.** We compute the penetration loss $\mathcal{L}_{\text{pen}}$ for all object-human pairs. We precompute a signed distance field for each object input and use it to compute the penetration depth between vertices of a human body and an object surface. This follows established practice in human-object penetration loss for interactions (Li & Dai, 2024; Hassan et al., 2019).

**Temporal Smoothness Loss.** We regularize the object motions $\{(\mathbf{R}_t, \mathbf{t}_t)\}_{t=1}^T$ to be temporally smooth based on the motion state attributes $(a_r, a_\tau)$ of each virtual object node. We thus compute

$$\mathcal{L}_{\text{r}} = \sum_{\mathcal{O}} \sum_t \begin{cases} \text{GD}(\mathbf{R}_t, \frac{1}{2}(\mathbf{R}_{t-1} + \mathbf{R}_{t+1})), & \text{if } a_r = \text{true} \\ \text{GD}(\mathbf{R}_t, \mathbf{R}_{t+1}), & \text{otherwise} \end{cases} \tag{5}$$

where $\text{GD}(\cdot)$ denotes the geodesic distance between two rotations. The top case, where spherical linear interpolation is used, promotes smooth rotational motions for object $\mathcal{O}$, while the bottom case penalizes temporal changes in object rotations. For the translations, we compute

$$\mathcal{L}_{\tau} = \sum_{\mathcal{O}} \sum_t \begin{cases} \mathcal{L}_2(\mathbf{t}_t, \frac{1}{2}(\mathbf{t}_{t-1} + \mathbf{t}_{t+1})), & \text{if } a_\tau = \text{true} \\ \mathcal{L}_2(\mathbf{t}_t, \mathbf{t}_{t+1}), & \text{otherwise} \end{cases} \tag{6}$$

where the top promotes smooth translational motions for object $\mathcal{O}$, while the bottom penalizes temporal changes in object translations. Overall, the temporal smoothness loss is $\mathcal{L}_{\text{smo}} = \mathcal{L}_{\text{r}} + \mathcal{L}_{\tau}$.

**Total Loss.** Our total loss is a weighted sum of the fitting loss, contact loss, penetration loss, and temporal smoothness loss:

$$\mathcal{L}_{\text{total}} = \lambda_{\text{fit}}\mathcal{L}_{\text{fit}} + \lambda_{\text{con}}\mathcal{L}_{\text{con}} + \lambda_{\text{pen}}\mathcal{L}_{\text{pen}} + \lambda_{\text{smo}}\mathcal{L}_{\text{smo}}. \tag{7}$$

## 4 EXPERIMENTS

We evaluate HOI-PAGE both qualitatively and quantitatively in diverse interaction scenarios, including single-person single-object, multi-person single-object, and single-person multi-object interactions. We show that our approach achieves superior generation realism, diversity, and text alignment when compared to the state-of-the-art methods (Peng et al., 2025; Li et al., 2024b)

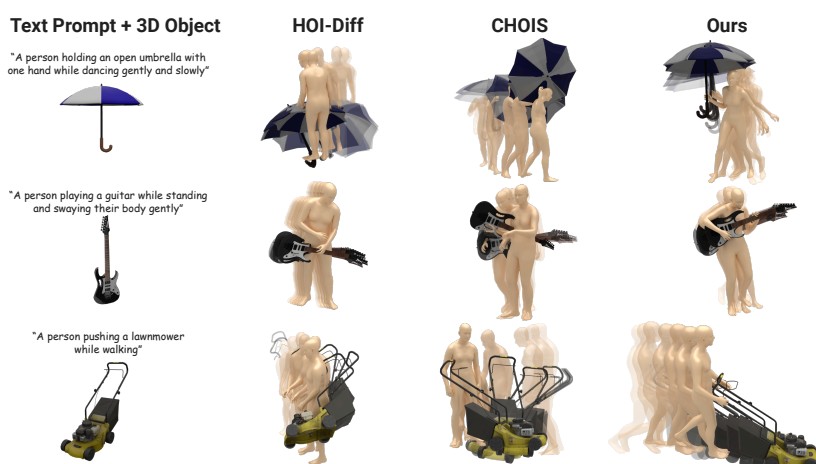

Figure 4: Single-person single-object interaction generations on the Sketchfab dataset. Our part affordance-guided approach generates more realistic 3D interaction motions with better text prompt alignment, compared to the baselines HOI-Diff and CHOIS, which struggle to generalize across diverse 3D objects (*e.g.*, lawnmower) unseen during training.

Table 1: Comparing single-person single-object interaction generations on the Sketchfab dataset. Our part affordance-guided approach generates realistic human-object interaction motions with semantic consistency, temporal smoothness, motion diversity, and physical plausibility metrics outperforming the baselines HOI-Diff and CHOIS that require 4D interaction data for supervision.

| | Semantics | Temporal Smoothness | | Motion Diversity | | Physical Plausibility | |
| | VideoCLIP ↑ | Human ↓ | Object ↓ | Human ↑ | Object ↑ | Non-collision ↑ | Contact ↑ |
|---|---|---|---|---|---|---|---|
| HOI-Diff | 0.233 | **0.007** | 0.035 | 0.35 | 0.72 | 0.98 | 0.76 |
| CHOIS | 0.239 | 0.009 | 0.009 | 0.44 | 0.49 | 0.98 | 0.64 |
| Ours | **0.250** | 0.008 | **0.006** | **0.47** | **0.80** | **0.99** | **0.92** |

**Dataset.** We collected 24 daily objects from Sketchfab.com, spanning categories such as household items, sports equipment, instruments, and transportation devices. Each object is a textured 3D mesh and canonicalized with a consistent upright orientation. A signed distance field (SDF) is precomputed for each object. We prepared 16 text prompts for single-person single-object interactions and 5 prompts for multi-person or multi-object scenarios, respectively.

**Baselines.** We compare with the state-of-the-art methods HOI-Diff (Peng et al., 2025) and CHOIS (Li et al., 2024b), which generate *single-person single-object interactions* from text prompts. These baselines were trained on real-world captured data of people interacting with indoor objects. We use the pre-trained models released by the authors and adapt them to the Sketchfab dataset, as we do not have any 4D ground truth for this data for training. CHOIS additionally requires object waypoints as input, which we provide by using the object waypoints generated by our approach.

**Evaluation Metrics.**
*- Perceptual Study.* We evaluate the realism and text alignment of 4D HOI motions. In a binary study, participants are shown two rendered interaction videos and asked to select the more realistic one and the one better matching a given text prompt, respectively. In a unary study, they are shown a single interaction video and asked to rate its realism and text alignment, respectively, from 1 (= strongly disagree) to 5 (= strongly agree). We surveyed 30 participants.
*- Semantic Alignment.* To measure alignment between a 4D HOI and a text prompt, we compute the cosine similarity between the text and the rendered video embeddings. A pre-trained VideoCLIP model (Bolya et al., 2025) (PE-Core-G14-448) is used to extract the embeddings. We render a 4D interaction from 3 different views and compute the average cosine similarity as the score.
*- Temporal Smoothness.* We evaluate the temporal smoothness of a generated 4D human motion by computing the distance between each 3D joint position at a given frame and the average position of the same joint in the two neighboring frames (similar to Eq. (6)-top). Similarly, the temporal smoothness of a 4D object motion is computed using the object's bounding box corners.
*- Motion Diversity.* To evaluate human motion diversity, we generate 5 interaction samples for each text prompt and compute the distance between each pair of samples for every joint position at a given frame. Object motion diversity is evaluated in the same way w.r.t. bounding box corners.

**Multi-person Single-Object Interaction Generation**  **Single-Person Multi-object Interaction Generation**

Figure 6: Our multi-person single-object and single-person multi-object interaction generations on the Sketchfab dataset. The flexibility of part affordable graphs enables our approach to generate diverse 3D interactions with multiple persons/objects.

Table 2: Multi-person single-object (MPSO) and single-person multi-object (SPMO) interaction generations on the Sketchfab dataset. Our approach handles well complex interaction scenarios involving multiple persons/objects, owing to the flexibility of our part affordance graphs, while achieving consistent performance in the perceptual ratings (on a scale of 1-5) and evaluation metrics.

| | Perceptual | | Semantics | Temporal Smoothness | | Motion Diversity | | Physical Plausibility | |
|---|---|---|---|---|---|---|---|---|---|
| | Realism↑ | Text Match↑ | VideoCLIP↑ | Human↓ | Object↓ | Human↑ | Object↑ | Non-collision↑ | Contact↑ |
| MPSO | 4.17 | 4.46 | 0.312 | 0.009 | 0.002 | 0.43 | 0.79 | 0.99 | 0.62 |
| SPMO | 4.46 | 4.59 | 0.268 | 0.005 | 0.005 | 0.54 | 0.87 | 0.99 | 0.90 |

- *Physical Plausibility (Non-collision, Contact)*. We compute non-collision and contact scores of a generated 4D interaction. At each frame, we check for collisions by querying each object's SDF for all human body vertices (Zhao et al., 2022). The non-collision score is defined as the ratio of the number of non-colliding human body vertices to the total number of vertices at each frame. The contact score is computed as the ratio of the number of frames with collision to the sequence length.

## 4.1 COMPARISON TO BASELINES

**Quantitative Evaluation.** The perceptual study results are shown in Fig. 5. In the binary evaluation, our 4D interaction generations are strongly preferred over HOI-Diff and CHOIS, receiving more than 91% of the votes for both realism and text alignment. In the unary evaluation, participants rated our generations with an average score of ∼4 for both criteria, significantly higher than HOI-Diff and CHOIS, which scored below 2. In Tab. 1, our approach achieves the best scores in semantic alignment, temporal smoothness of object motions, motion diversity, and physical plausibility metrics. HOI-Diff has slightly better temporal smoothness for human motions, but its generations do not align well with the text prompts and have the lowest human motion diversity. In contrast, our approach generates more diverse human motions. The perceptual studies and quantitative results show that our part-level contact distillation from LLMs is effective in generating more realistic and text-aligned interactions.

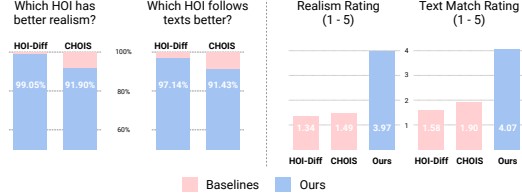

Figure 5: Perceptual studies of single-person single-object interaction generations on the Sketchfab dataset. In the binary study (**left**), participants strongly prefer our method over the baselines HOI-Diff and CHOIS for interaction realism and text matching. In the unary study (**right**), our generations achieve the highest ratings (on a scale of 1-5) compared to the baselines.

**Qualitative Evaluation.** Fig. 4 presents comparisons of generated 4D interactions. HOI-Diff and CHOIS struggle to generate plausible interactions for the Sketchfab objects unseen during their training. For example, HOI-Diff produces nearly static human poses with the guitar and has significant penetration with the lawnmower, while CHOIS generates less precise part-level contact between human hands and the lawnmower handle. In contrast, our approach generalizes better across different objects in zero shot, capturing well part-level affordances of objects.

## 4.2 MULTI-INTERACTION EVALUATION

In contrast to fully-supervised baselines that require real-world 4D captures for training (Peng et al., 2025; Li et al., 2024b), our zero-shot part-guided approach enables synthesizing more general, complex interaction scenarios, such as multi-person single-object generation and single-person multi-object generation. Fig. 6 shows our approach on these multi-interaction scenarios, by simply distill-

Table 3: Ablation studies on Sketchfab. Results are averaged over multi-person single-object and single-person multi-object interaction generations. Object motion smoothness, diversity, and physical contact scores degrade significantly without part-level fitting (PF), part-level contact (PC), and object motion states (OMS) constraints from part affordance graphs.

| | VideoCLIP↑ | Smoothness↓ | Diversity↑ | Non-collision↑ | Contact↑ |
|---|---|---|---|---|---|
| w/o PF | **0.290** | **0.004** | 0.81 | 0.99 | **0.76** |
| w/o PC | 0.289 | 0.011 | 0.71 | **1.00** | 0.26 |
| w/o OMS | **0.290** | 0.006 | 0.78 | 0.99 | 0.73 |
| Ours | **0.290** | **0.004** | **0.83** | 0.99 | **0.76** |

ing multi-person or multi-object nodes and their corresponding part nodes from the LLM during PAG construction. We also quantitatively evaluate our multi-interaction generation in Tab. 2. Although contact can become more challenging with the multi-person scenario, with more human contact constraints to satisfy, our approach synthesizes interaction sequences of quality that closely matches the simpler single-person single-object interactions in these more complex interaction scenarios. More results on multi-person multi-object generation are provided in the appendix.

## 4.3 ABLATION STUDIES

Fig. 7 and Tab. 3 show the results of our ablation studies on the Sketchfab dataset. We evaluate the effectiveness of our part affordance graph constraints: part-level fitting (*i.e.*, $\mathcal{L}_{3D}^{o}, \mathcal{L}_{2D}^{o}$), part-level contact (*i.e.*, $\mathcal{L}_{cc}$), and object motion states (*i.e.*, $a_r, a_\tau$ in $\mathcal{L}_{smo}$).

**What is the impact of part-level fitting?** Our part-level fitting (PF) during HOI optimization is essential for higher-level semantic plausbility not easily captured by standard quantitative metrics. Note that contact is measured at the whole body level, as we lack ground truth for part contacts. For instance, as shown in Fig. 7 (left), without part fitting, the ironing board has a wrongly tilted upwards orientation and significant motion, while using part fitting provides more meaningful semantic coherence.

**How do part contact constraints influence interaction quality?** Without part-level contact constraints (w/o PC), high-level motions are plausible but miss important contacts, our part contact constraints enable grasping of the iron handle with the person's hand in Fig. 7 (left middle).

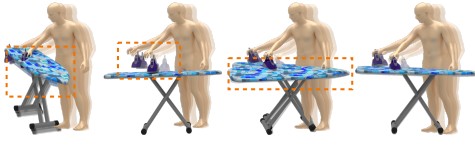

w/o Part Fitting    w/o Part Contact    w/o Object Motion States    Ours

Figure 7: Visualization of ablation studies on part affordance graph constraints. Without part-level fitting, the ironing board orientation is incorrect (tilted up); without part-level contact, the hand is not in contact with the iron's handle; without object motion states, the ironing board does not remain stationary. Using all part affordance graph constraints produces the most realistic interactions.

**What is the effect of characterizing object motion states?** Our characterization of object motion (OMS) in the PAG produces more semantically plausible object motion; for instance, this helps the ironing board remain stationary in Fig. 7.

**Limitations.** Capturing detailed, nuanced motion (e.g., individual finger articulations) remains a challenge, lying beyond the granularity of our PAGs, which could be addressed through physics-based simulation. Additionally, while we maintain robustness to image and video synthesis variations, synthesis will suffer if none of the generated first frames or videos produce satisfactory results.

## 5 CONCLUSION

We presented a new approach for zero-shot 4D human-object interaction synthesis that moves beyond whole-body interaction modeling by explicitly incorporating part-level affordances. By introducing part affordance graphs, and using them to guide video motion generation as well as 4D HOI optimization, our method enables more realistic, diverse, and generalizable interactions across a wide range of objects and scenarios, including complex multi-object and multi-person interactions. We hope this step towards finer-grained understanding of interactions in a zero-shot fashion will opening new possibilities in graphics and content creation, as well as in applications such as robotics and embodied AI.

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

# APPENDIX

In this appendix, we provide additional results in Appendix A and more implementation details in Appendix B.

## A  ADDITIONAL RESULTS

**Multi-person Multi-object Interaction Generation.** Fig. 8-left demonstrates that our part affordance graph-based approach is flexible and can generate more complex multi-person multi-object interactions. Fig. 8-right shows that our approach can generate interactions involving more than 2 people in a zero-shot fashion, going well beyond the single-person single-object interaction generation setting focused in existing works (Peng et al., 2025; Li et al., 2024b).

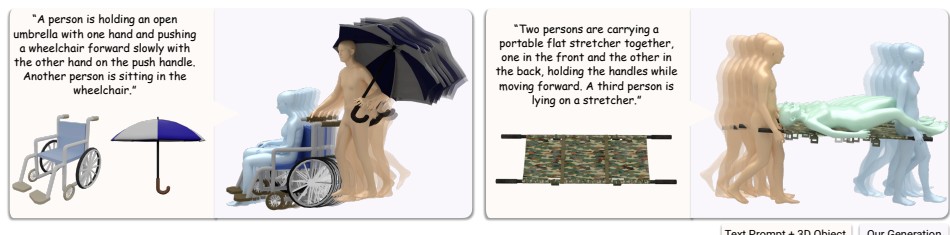

Figure 8: Our approach can generate multi-person multi-object interactions (**Left**) as well as interactions involving more than 2 people (**Right**).

**Diversity Visualization.** We visualize the generation diversity of our approach in Fig. 9. Given the same text prompt and 3D objects, our approach generates diverse 4D HOI interaction motions by varying the random noise used in video diffusion.

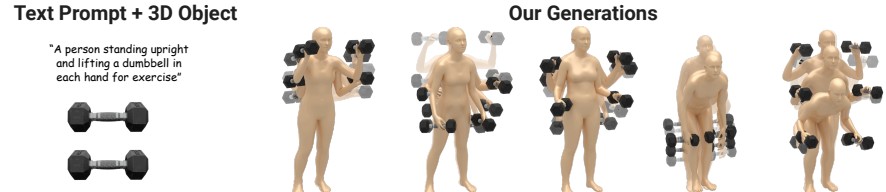

Figure 9: Our approach generates diverse 4D human-object interaction motions given the same text prompt and 3D objects as input.

**Intermediate Result Visualization.** Fig. 10 presents intermediate results at different stages of our pipeline, including inferred part affordance graphs (Sec. 3.2), enhanced text prompts, 3D object part segmentation (Sec. 3.3), interaction video generation, video object segmentation, depth estimation, and human motion recovery (Sec. 3.4).

**Ablating Foundation Models.** We test our pipeline with different Large Language Models (LLMs) and Video Diffusion Models (VDMs). Tab. 4 shows the ablation results on the Sketchfab dataset with averaged performance over multi-person single-object and single-person multi-object interaction generations. Our original pipeline uses DeepSeek as the LLM and CogVideoX as the VDM. The first row reports the performance of using Gemini as the LLM, while the second row reports the performance of using Hunyuan-Video as the VDM. We observe that our approach achieves stable performance across the foundation models used.

Table 4: Evaluating different Large Language Models (LLMs) and Video Diffusion Models (VDMs) on Sketchfab. The performance of our approach (based on DeepSeek and CogVideoX) remains stable when using a different LLM (Gemini) or VDM (HunyuanVideo). Results are averaged over multi-person single-object and single-person multi-object interaction generations.

|  | VideoCLIP↑ | Smoothness↓ | Diversity↑ | Non-collision↑ | Contact↑ |
|---|---|---|---|---|---|
| LLM | 0.291 | 0.004 | 0.73 | 0.99 | 0.68 |
| VDM | 0.289 | 0.002 | 0.81 | 0.99 | 0.76 |
| Ours | 0.290 | 0.004 | 0.83 | 0.99 | 0.76 |

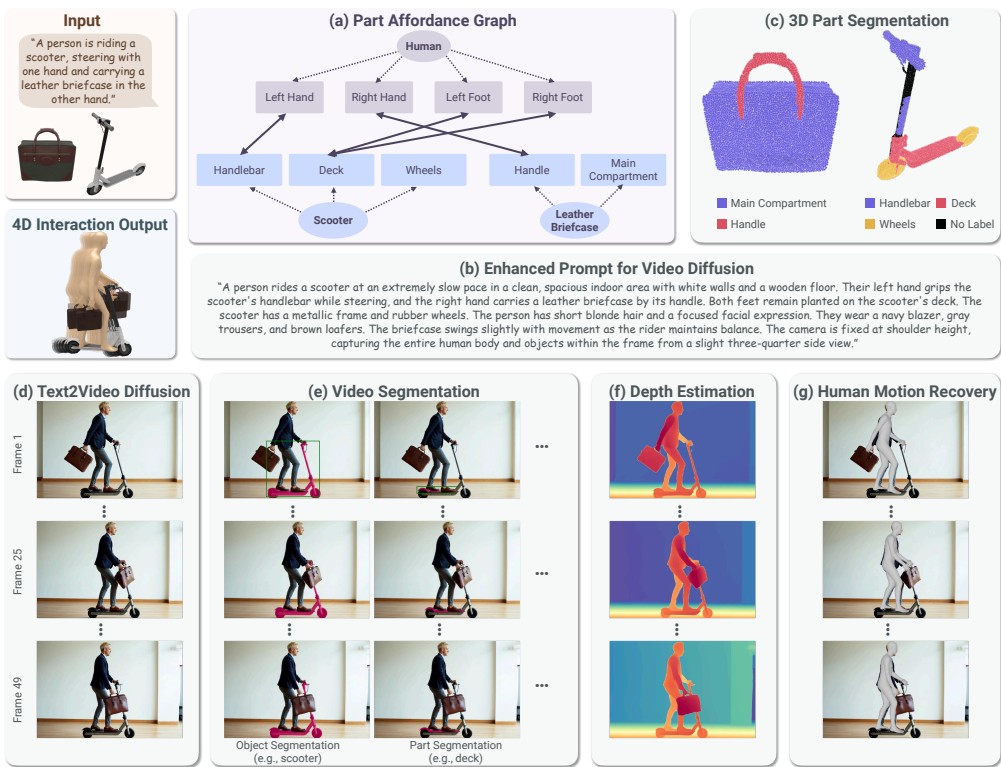

Figure 10: Intermediate result visualization. Given 3D objects (e.g., a leather briefcase and a scooter) and a text prompt, we use an LLM to infer the part affordance graph (a). We also use the LLM to perform prompt enhancement (b) to capture the interaction details in (a) for video generation. We perform multi-view part segmentation (c) on the input 3D objects based on (a). Next, we generate an interaction video (d) guided by (b). We then detect, track, and segment objects and their parts in the video (e), estimate depth for each frame (f), and perform human motion recovery to estimate 4D human poses from the video.

**Evaluation on the BEHAVE Dataset.** We further evaluate our approach on the BEHAVE dataset (Bhatnagar et al., 2022), which contains real-world object scans and HOI captures. BEHAVE's test set has 18 objects. We sample 3 text prompts for each object and generate 5 interaction variations for each prompt.

We use the released models of HOI-Diff (Peng et al., 2025) and CHOIS (Li et al., 2024b) for comparison. Note that HOI-Diff was trained on BEHAVE, while CHOIS was trained on the FullBody-Manipulation dataset (Li et al., 2023), which contains indoor object interaction captures similar to BEHAVE. CHOIS is conditioned additionally on object waypoints, which are derived from our generation results. The same set of metrics from Sec. 4.1, including semantic alignment, temporal smoothness, motion diversity, and physical plausibility, are computed.

Tab. 5 presents the quantitative comparisons, where our approach performs better than HOI-Diff and CHOIS in terms of semantic alignment, temporal smoothness, motion diversity, and physical contact. Fig. 11 shows the qualitative comparisons. Our approach synthesizes 4D interactions aligned more closely with text prompts than the generations of HOI-Diff and CHOIS, which require captured interaction data for supervision.

## B  IMPLEMENTATION DETAILS

Our HOI-PAGE is implemented using PyTorch (Paszke et al., 2019). To improve realism of a synthesized HOI video (Sec. 3.3), we generate 5 candidate images for the first frame using FLUX and then select the one with the best visual quality w.r.t. human anatomy, text alignment, and camera views by querying a VLM (GPT-4.1). We use 50 denoising steps for both image and video diffusion.

Table 5: Comparing single-person single-object interaction generations on the BEHAVE dataset. Our approach achieves better performance than HOI-Diff (Peng et al., 2025) and CHOIS (Li et al., 2024b) in semantic consistency, temporal smoothness, motion diversity, and physical contact metrics.

| | Semantics | Temporal Smoothness | | Motion Diversity | | Physical Plausibility | |
|---|---|---|---|---|---|---|---|
| | VideoCLIP ↑ | Human ↓ | Object ↓ | Human ↑ | Object ↑ | Non-collision ↑ | Contact ↑ |
| HOI-Diff | 0.200 | 0.007 | 0.015 | 0.34 | 0.54 | **0.99** | 0.72 |
| CHOIS | 0.214 | 0.009 | 0.008 | 0.48 | 0.47 | 0.98 | 0.61 |
| Ours | **0.220** | **0.006** | **0.004** | **0.61** | **0.92** | 0.98 | **0.78** |

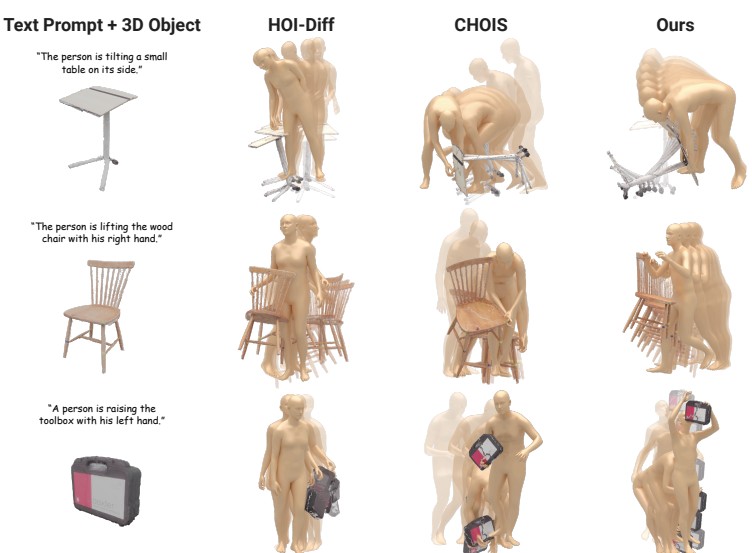

Figure 11: Qualitative comparisons of single-person single-object interaction generations on the BEHAVE dataset. Given real-world object scans and text prompts from BEHAVE, our 4D interaction generations align more closely with the text input than those of HOI-Diff (Peng et al., 2025) and CHOIS (Li et al., 2024b), which are specifically trained on captured data of real people interacting with such objects.

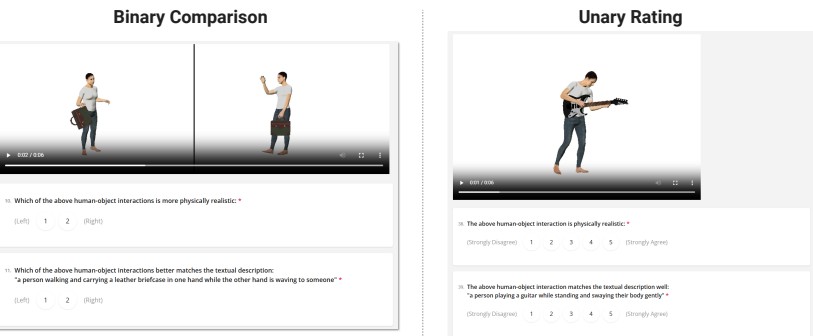

Figure 12: Screenshots of our perceptual study survey. Binary study (**Left**): participants are asked to select a 4D interaction generation with better realism and text alignment, respectively. Unary study (**Right**): rate generation realism and text alignment, respectively, on a scale from 1 to 5.

CogVideoX generates 49 frames per video, and thus $T = 49$. We optimize $\mathcal{L}_{\text{total}}$ for 600 steps using gradient descent, which takes ~6 mins for single-object interactions and ~10 mins for interactions involving 2 objects on A100 GPUs. We repeat the optimization for 4 times with different sampled object rotation initializations around the up axis to mitigate convergence to local optimum caused by Chamfer Distance in $\mathcal{L}_{\text{fit}}$. Prompts for part affordance graph inference with LLMs and first-frame selection with VLMs are provided below.

**Point Map Alignment.** To estimate point maps (or depth) for the generated video frames, we use MoGe (Wang et al., 2024) due to its strong generalization to open-domain images and its more regu-

larized 3D structure estimation (Sec. 3.3). However, MoGe is a single-image estimation method and suffers from inconsistencies across video frames. Its point map estimation also does not align well with the 4D human motion estimated by GVHMR (Shen et al., 2024). To address this, we perform a point map alignment step, leveraging the recovered 4D human motion as guidance. We first detect and segment humans in the generated video frames, similar to Video Object Part Segmentation in (Sec. 3.3). We then optimize the scale, rotation, and translation of each point map frame so that the human point maps are aligned with the 4D human motion. The optimization objective combines 3D and 2D fitting losses based on Chamfer distance, similar to $\mathcal{L}_{3D}^{\mathcal{O}}$ and $\mathcal{L}_{2D}^{\mathcal{O}}$ in Sec. 3.4. We perform 300 steps of gradient descent for this optimization.

**Perceptual Study.** In our binary perceptual study, we have 14 generation comparisons, where each comparison consists of two questions: one for realism and one for text alignment. In the unary study, participants are asked to rate 31 generations on realism and text alignment, respectively. Fig. 12 shows the screenshots of our perceptual study survey.

**Prompting for Part Affordance Graph Inference.** We provide the text prompt below for instructing an LLM (Guo et al., 2025) to infer part affordance graphs (Sec. 3.1), while simultaneously enhancing short interaction prompts into longer, more detailed ones.

```
You are a helpful assistant in analyzing human-object interactions.

- Task: You will be given a list of objects and a short text description
    of human interactions with these objects. Your task is to analyze all
     the interaction relations among human body parts and object parts
    and output the results as a graph in the JSON format.

- Input format: The input is provided in the JSON format as follows
{
        "objects": [
                "object 1",
                "object 2"
        ],
        "interaction": "a short interaction description"
}

- Output format: Provide the output strictly in JSON format, without any
    additional explanation or commentary, structured as follows:
{
        "object part nodes": [
                "object 1, object part 1",
                "object 1, object part 2"
        ],
        "body part nodes": [
                "person 1, human body part 1",
                "person 1, human body part 2"
        ],
        "interaction edges": [
                {
                        "nodes": [
                                "object a, object part b",
                                "person c, human body part d"
                        ],
                        "is_rel_static": <true or false indicating if the
                                two nodes' movements remain relatively
                                stationary during interaction>,
                        "is_continuous": <true or false indicating if the
                                two nodes remain in continuous physical
                                contact during interaction>
                },
                {
                        "nodes": [
                                "object x, object part y",
                                "person z, human body part w"
                        ],
```

```
918                               "is_rel_static": <true or false>,
919                               "is_continuous": <true or false>
920                           }
921             ],
922             "interaction": "a long description in 150 words summarizing the
923                 output interaction graph to guide a realistic video
                    generation",
924             "object states": [
925                     {
926                               "name": "object 1",
927                               "is_translational": <true or false indicating if
928                                   object 1 has translational motions during
                                      interaction>,
929                               "is_rotational": <true or false indicating if
930                                   object 1 has rotational motions during
931                                   interaction>,
932                               "description": "a short description in 20 words
933                                   identifying object 1 during interaction"
                      },
934                     {
935                               "name": "object 2",
936                               "is_translational": <true or false>,
937                               "is_rotational": <true or false>,
938                               "description": "a short description in 20 words
939                                   identifying object 2 during interaction"
                      }
940             ],
941             "human states": [
942                     {
943                               "name": "person 1",
944                               "description": "a short description in 20 words
945                                   identifying person 1 during interaction"
                      }
946             ]
947 }
948
949 - Rules for analysis:
      (1) There are two types of nodes in the output interaction graph: "
950       object part nodes" representing object parts and "body part nodes"
951       representing human body parts.
952   (2) The "object part nodes" field represent a part-level segmentation
953       of each input object. Segmentations should roughly cover the entire
954        object without becoming excessively detailed. Use descriptive,
955       specific part names rather than generic terms, for example, avoid "
956       surface", "edge", "body", "base", "area", "cover", "support", "
957       connector", "frame", and the like. Do not differentiate between
958       left and right parts. Avoid numbering object parts. Example: For a
959       "bike", use the following parts: "handlebar", "pedal", "seat", "
960       frame tubes", "wheels". For a "skateboard", use the following parts
961       : "longboard deck", "wheels". For a "cordless vacuum cleaner", use
962       the following parts: "ergonomic hand grip", "wand", "floor roller".
963        For a "ladder", use the following parts: "side rail tubes", "rungs
964       ". For a "boxing bag", use the following parts: "punching bag".
965   (3) The "body part nodes" field must be the following: "left hand", "
966       right hand", "left arm", "right arm", "left shoulder", "right
967       shoulder", "left leg", "right leg", "left foot", "right foot", "
968       head", "hips". Distinguish between left/right human body parts.
969   (4) The "interaction edges" represent direct physical contact
970       relationships between two end nodes. An edge connects an object
971       part node to either a human body part node or another object part
          node. Do not connect part nodes within the same object. Example:
          when ironing on an ironing board, the soleplate part of an iron
          should be connected to the top flat panel part of the ironing board
          . Each edge has two attributes: "is_continuous" and "is_rel_static
          ". The "is_continuous" attribute is true if the two end nodes are
```

in continuous physical contact during the interaction process, otherwise false. Example: when holding a dumbbell, the hand is in continuous contact with the handle without any separation; when punching a boxing bag, the hands are not in continuous contact with the bag; when a person stepping up a ladder, the feet and hands are both not in continuous contact with the rungs. The "is_rel_static" attribute is true if the two end nodes' movements are relatively stationary to each other while being in continuous physical contact during the interaction process, otherwise false. Example: when riding a bike, hands are relatively stationary to the handlebar; when playing a guitar, the hand strumming strings is not relatively stationary to the main compartment of the guitar.

(5) Explicitly mentioned body parts in the input "interaction" field must be included. Example: For a description "a person is lifting a single dumbbell with one hand", include either "left hand" or "right hand" in the analysis. If no specific body part is mentioned, use the most common ergonomic interactions in the physical contact analysis.

(6) Focus on primary actions influencing object use or movement in the physical contact analysis. Example: For "a person walking and carrying a briefcase in one hand", the primary action for analysis is "carrying".

(7) Ensure the identified object parts belong to their respective objects in the node and edge outputs of the interaction graph.

(8) Ensure plausible distribution and avoid conflicts or duplication of human body parts during the interaction analysis.

(9) Exclude environmental elements, like floor, ground, or wall, from the physical contact analysis.

(10) The "interaction" field in the output JSON must concisely summarize the "interaction edges" of the graph to guide realistic video generation. Follow this structure:

   (a) Begin with the interaction(s) as described in the input short "interaction" description. Clearly specify each participant's role if multiple people or objects are involved. All motions must occur at an extremely slow pace.

   (b) Then describe the interaction motion details, focusing on physical contact between human body parts and object parts. If a human is specified to be non-static, make sure their body parts without physical contact show expressive movement. For example, when "skateboarding", the person's arms can swing to maintain balance, and the legs can bend slightly; when "cleaning with a cordless vacuum cleaner", the arm that is not holding the vacuum can swing naturally while walking; when "riding a scooter", one foot can remain static on the deck while the other swings to push off the ground and gain speed. Importantly, the human body parts without physical contact must also move in slow motion.

   (c) Next, describe the appearance of people, objects, and environments. For people, you must strictly include the following four aspects: their hair styles, facial expressions, clothes, and shoes. For example, "short black hair", "neutral facial expression", "wearing a gray shirt, blue jeans, and white sneakers". For objects, describe general type and appearance without overly specific details. The environment is always a clean, spacious indoor area with white walls and a wooden floor. Ensure the environment supports the action without adding unnecessary complexity.

   (d) The "interaction" summarization must not exceed 150 words.

(11) The "object states" in the output JSON have four attributes, "name", "is_translational", "is_rotational", and "description", for each object. The "is_translational" attribute is true if the corresponding object has global translational motions during interaction, otherwise false. The "is_rotational" attribute is true if the corresponding object has global rotational motions during interaction, otherwise false. Both "is_translational" and "

```
      is_rotational" attributes must consider only the object's overall
      motion, not motions of individual parts, for example, a bike being
      ridden should be considered as moving translationally as a whole,
      while ignoring the rotation of its pedals. The object "description"
       attribute should clearly identify the object by briefly stating
      its type, appearance, and its interactions with human bodies, using
       no more than 20 words. The object "description" should be based on
       relevant "interaction edges" and the long "interaction" fields in
      the output. In the object "description", avoid using numerical or
      ordinal references.
   (12) The "human states" in the output JSON have two attributes, "name"
      and "description", for each person. The human "description"
      attribute should clearly identify the person by briefly stating
      their appearance and interactions with object parts in 20 words.
      The human "description" should be based on relevant "interaction
      edges" and the long "interaction" fields in the output. Avoid using
       numerical or ordinal references in the "description" attribute.

- Examples:
  (1) If the input is
{
        "objects": [
                "umbrella",
                "suitcase"
        ],
        "interaction": "a person is dragging a suitcase with one hand and
            holding an open umbrella with the other hand while walking"
}
  then the output is
{
        "object part nodes": [
                "umbrella, canopy",
                "umbrella, shaft",
                "suitcase, main compartment",
                "suitcase, handle",
                "suitcase, wheels"
        ],
        "body part nodes": [
                "person 1, left hand",
                "person 1, right hand",
                "person 1, left arm",
                "person 1, right arm",
                "person 1, left shoulder",
                "person 1, right shoulder",
                "person 1, left leg",
                "person 1, right leg",
                "person 1, left foot",
                "person 1, right foot",
                "person 1, head",
                "person 1, hips"
        ],
        "interaction edges": [
                {
                        "nodes": [
                                "umbrella, shaft",
                                "person 1, left hand"
                        ],
                        "is_rel_static": true,
                        "is_continuous": true
                },
                {
                        "nodes": [
                                "suitcase, handle",
                                "person 1, right hand"
                        ],
```

```
1080                                   "is_rel_static": true,
1081                                   "is_continuous": true
1082                               }
1083                           ],
1084                      "interaction": "A person is dragging a suitcase's handle with the
1085                          right hand and holding a open umbrella's shaft with the left
1086                          hand while walking at a slow pace. The suitcase rolls
1087                          smoothly behind them as they move, and the open umbrella is
1088                          held steadily above. The person has black short hair and a
1089                          neutral facial expression. They wear a gray shirt, blue jeans
1090                          , and white sneakers. The scene takes place in a clean,
1091                          spacious indoor area with white walls and a wooden floor.",
1092                      "object states": [
1093                              {
1094                                   "name": "umbrella",
1095                                   "is_translational": true,
1096                                   "is_rotational": false,
1097                                   "description": "the open umbrella being held"
1098                              },
1099                              {
1100                                   "name": "suitcase",
1101                                   "is_translational": true,
1102                                   "is_rotational": false,
1103                                   "description": "the suitcase being dragged"
1104                              }
1105                          ],
1106                      "human states": [
1107                              {
1108                                   "name": "person 1",
1109                                   "description": "the person with black short hair
1110                                       who is wearing gray shirt and blue jeans and
1111                                       holding/dragging the objects"
1112                              }
1113                          ]
1114    }

     (2) If the input is
     {
             "objects": [
                     "bike"
             ],
             "interaction": "a person is riding a bike"
     }
     then the output is
     {
             "object part nodes": [
                     "bike, handlebar",
                     "bike, pedal",
                     "bike, seat",
                     "bike, frame tubes",
                     "bike, wheels"
             ],
             "body part nodes": [
                     "person 1, left hand",
                     "person 1, right hand",
                     "person 1, left arm",
                     "person 1, right arm",
                     "person 1, left shoulder",
                     "person 1, right shoulder",
                     "person 1, left leg",
                     "person 1, right leg",
                     "person 1, left foot",
                     "person 1, right foot",
                     "person 1, head",
                     "person 1, hips"
```

```
1134              ],
1135              "interaction edges": [
1136                      {
1137                              "nodes": [
1138                                      "bike, handlebar",
1139                                      "person 1, left hand"
1140                              ],
1141                              "is_rel_static": true,
1142                              "is_continuous": true
1143                      },
1144                      {
1145                              "nodes": [
1146                                      "bike, handlebar",
1147                                      "person 1, right hand"
1148                              ],
1149                              "is_rel_static": true,
1150                              "is_continuous": true
1151                      },
1152                      {
1153                              "nodes": [
1154                                      "bike, pedal",
1155                                      "person 1, left foot"
1156                              ],
1157                              "is_rel_static": true,
1158                              "is_continuous": true
1159                      },
1160                      {
1161                              "nodes": [
1162                                      "bike, pedal",
1163                                      "person 1, right foot"
1164                              ],
1165                              "is_rel_static": true,
1166                              "is_continuous": true
1167                      },
1168                      {
1169                              "nodes": [
1170                                      "bike, seat",
1171                                      "person 1, hips"
1172                              ],
1173                              "is_rel_static": true,
1174                              "is_continuous": true
1175                      }
1176              ],
1177              "interaction": "A person is riding a bike at a slow, steady pace
1178                  in a clean, spacious indoor area with white walls and a
1179                  wooden floor. Their hands grip the handlebars firmly and feet
1180                   remain securely on the pedals. The bike has a simple, modern
1181                   design with a black frame and straight handlebars. The rider
1182                  has short brown hair and a neutral facial expression. They
1183                  wear a blue shirt, black shorts, and white sneakers.",
1184              "object states": [
1185                      {
1186                              "name": "bike",
1187                              "is_translational": true,
                                "is_rotational": false,
                                "description": "the bike having a black frame and
                                    being ridden"
                        }
                ],
                "human states": [
                        {
                                "name": "person 1",
                                "description": "the person who is wearing blue
                                    shirt and black shorts and riding"
                        }
```

```
1188              ]
1189     }
1190
1191        (3) If the input is
1192     {
1193              "objects": [
1194                       "guitar"
1195              ],
1196              "interaction": "a person is playing a guitar while standing"
1197     }
1198        then the output is
1199     {
1200              "object part nodes": [
1201                       "guitar, neck",
1202                       "guitar, main compartment"
1203              ],
1204              "body part nodes": [
1205                       "person 1, left hand",
1206                       "person 1, right hand",
1207                       "person 1, left arm",
1208                       "person 1, right arm",
1209                       "person 1, left shoulder",
1210                       "person 1, right shoulder",
1211                       "person 1, left leg",
1212                       "person 1, right leg",
1213                       "person 1, left foot",
1214                       "person 1, right foot",
1215                       "person 1, head",
1216                       "person 1, hips"
1217              ],
1218              "interaction edges": [
1219                       {
1220                                "nodes": [
1221                                         "guitar, neck",
1222                                         "person 1, left hand"
1223                                ],
1224                                "is_rel_static": false,
1225                                "is_continuous": true
1226                       },
1227                       {
1228                                "nodes": [
1229                                         "guitar, main compartment",
1230                                         "person 1, right hand"
1231                                ],
1232                                "is_rel_static": false,
1233                                "is_continuous": true
1234                       }
1235              ],
1236              "interaction": "A person is playing a guitar while standing in a
1237                  clean, spacious indoor area with white walls and a wooden
1238                  floor. Their left hand is holding the guitar's fretboard, and
1239                   their right hand is strumming the strings slowly. The guitar
1240                   is a classic acoustic model with a polished wood finish. The
1241                   person has short brown hair and a happy faical expression.
                     They wear a black shirt, blue jeans, and black boots, gently
                     swaying their body to the rhythm.",
                 "object states": [
                         {
                                  "name": "guitar",
                                  "is_translational": true,
                                  "is_rotational": false,
                                  "description": "the wooden guitar being played"
                         }
                 ],
                 "human states": [
```

```
            {
                    "name": "person 1",
                    "description": "the person with short brown hair
                        who is wearing blue jeans and playing the
                        guitar"
            }
        ]
}
```

**Prompting for First-Frame Selection.** The following text prompt is used to instruct a VLM (GPT-4.1) to select the best first frame from a candidate set (Appendix B) for video diffusion.

```
You are a helpful assistant in image understanding and comparison.
- Task: You will receive one image file that actually contains two
    separate images shown side-by-side (left and right), along with a
    short text describing human-object interactions. Look closely at both
     images and read the text description. Use the "Analysis Rules" below
     to decide which single image ("left" or "right") is a better match
    for both the rules and the text description.
- Input format:
        (1) One image file that includes two images placed next to each
            other horizontally, like this: [left image | right image].
        (2) One short text that describes the human-object interactions
            that should be happening in the images.
- Output format: You must output only one word: either "left" or "right".
    Do not add any other words, explanations, or comments.
- Analysis Rules:
        (1) Full Human Figures: Prefer the image where people are shown
            completely, from their heads down to their feet, inside the
            image area, and where the front faces of the main people
            involved in the interaction are clearly visible.
        (2) Correct Anatomy: Prefer the image where humans have normal-
            looking body parts and proportions. Avoid images showing
            people with distorted, disfigured, or anatomically incorrect
            limbs or bodies.
        (3) Matching Text Description: Prefer the image where the human-
            object interactions match the provided short text description
            .
        (4) Plausible Interactions: Prefer the image where interactions
            between people and objects look natural, physically plausible
            . Avoid interactions that involve problematic body parts,
            like strangely bent or extra limbs. Avoid images with
            unrealistic physics, like people or objects floating in the
            air.
        (5) Camera View: Prefer wide-shot images taken from a shoulder-
            height, three-quarter side view that clearly shows both the
            pose and the interaction. If that's not available, prefer
            side views over straight-on front views. Avoid images taken
            from high-up, low-down, or close-up views that crop or
            obscure full human figures. Also avoid images where people or
             objects are too close to walls or background objects.
        (6) Sharp Details: Prefer images with clear, sharp details, and
            avoid images with motion blur around human body parts.
        (7) Realistic Style: Prefer photographic or realistic images over
             cartoons, drawings, illustrations, or images with very
            artistic styles.
        (8) Do not consider the mood, feeling, or atmosphere of the image
             in your comparison.
```

**LLM Usage Disclosure.** LLMs (ChatGPT and Gemini) were used for correcting grammatical error and typos and finding synonyms in paper writing.

