# OpenReview forum: "HOI-PAGE: Zero-Shot Human-Object Interaction Generation with Part Affordance Guidance"
_ICLR.cc/2026/Conference — Submitted to ICLR 2026_

### Official Review · Reviewer_LToL · 2025-11-01

**Soundness:** 2
**Presentation:** 3
**Contribution:** 2
**Rating:** 4
**Confidence:** 5

**Summary:**

The paper proposes HOI-PAGE, a zero-shot framework for generating 4D human–object interactions (HOI) guided by part-level affordances. It introduces a Part Affordance Graph (PAG) that links human and object parts through semantic and geometric reasoning derived from large language models. The method works in three stages: Part segmentation of objects using multi-view visual-language models (Qwen-VL + SAM2). Text-to-video HOI generation with part-aware descriptions (via CogVideoX and FLUX). Part-affordance-guided optimization to reconstruct 4D motion, enforcing contact and physical plausibility. Without any 4D HOI training data, HOI-PAGE generalizes to unseen objects and complex multi-person scenes, showing better realism, text alignment, and contact accuracy than prior works like HOI-Diff and CHOIS.

**Strengths:**

- The method requires no 4D interaction training data, yet generalizes to unseen objects and multi-entity scenes, demonstrating impressive flexibility.
- Each stage (part segmentation, text-to-video, 4D fitting) is well-defined and independently improvable, making the system extensible as better diffusion or vision models appear.

**Weaknesses:**

- The paper lacks quantitative metrics to evaluate the accuracy of object motion and the penetration ratio in multi-object interaction scenarios.
- The method fails to generate fine-grained contacts such as realistic hand–object grasps. The hand poses often appear averaged or static, resembling a mean pose rather than physically adaptive interactions.

**Questions:**

- Although LLMs can infer affordance information in a zero-shot manner and often provide reasonable contact region predictions, it remains unclear how accurately the object’s part segmentation aligns with the regions specified by the LLM. The paper does not include any quantitative or qualitative validation of this alignment—only two example visualizations are shown in the Appendix, which are insufficient to verify the segmentation quality.

---

### Official Review · Reviewer_Hz1Q · 2025-11-01

**Soundness:** 2
**Presentation:** 3
**Contribution:** 2
**Rating:** 6
**Confidence:** 3

**Summary:**

The paper introduces HOI-Page, a novel method for zero-shot 4D generation of human-object interaction from text prompts and 3D objects. The core idea is a Part Affordance Graph (PAG) that, distilled from large language models, can capture the contact relationships between human body parts and object parts. The authors further introduce a three-stage generation pipeline based on PAG: (1) decomposing the 3D objects into geometric parts, (2) generating HOI videos from text, and (3) optimizing human and object motions based on the generated video and contact constraints. Extensive experiments have been conducted to demonstrate the performance of HOI-PAGE over existing methods.

**Strengths:**

+ The three-stage approach is logically sound. The results produced by HOI-PAGE indeed present improvements over existing methods, in terms of penetration.

+ The paper is well-written and easy to follow.

**Weaknesses:**

- Limited novelty. (1) The part-level affordance map is not novel. Existing methods have explored the usage of contact graph, affordance map, and video generation, etc. (2) The usage of LLMs is also not novel, which may limit the performance of the 4D generation as well.

- Limited granularity of motions. The model cannot address the penetration issue. Additionally, the generated motion may still present unwanted instability, especially for the objects.

- More evaluations may be needed. (1) Comparisons with methods like ZeroHSI, InterDreamer would benefit the evaluations. (2) Evaluations on the similarity between the generated video and the 4D HOI motion sequence would be required. (3) The correctness/stability/failure rate of the part affordance graph would be better to evaluate.

**Questions:**

Please see the weaknesses listed above.

---

### Official Review · Reviewer_fjwo · 2025-11-01

**Soundness:** 2
**Presentation:** 3
**Contribution:** 2
**Rating:** 4
**Confidence:** 4

**Summary:**

This paper presents HOI-PAGE, a new zero-shot approach for synthesizing 4D human-object interactions (HOIs) from text prompts based on part-level affordance reasoning. The authors argue that realistic HOIs require a fine-grained understanding of how human body parts engage with specific object parts, a detail missed by prior global-focused methods. Their core contribution is the Part Affordance Graph (PAG), a structured HOI representation distilled from Large Language Models (LLMs) that encodes these specific part-to-part contact relations. This PAG then guides a three-stage synthesis pipeline: first, decomposing the input 3D objects into geometric parts ; second, generating a reference HOI video from which motion constraints are extracted ; and finally, optimizing a 4D HOI motion sequence that both mimics the reference dynamics and, critically, satisfies the part-level contact constraints from the PAG. Experiments show this approach is flexible, capable of generating complex multi-person or multi-object interactions, and achieves significantly improved realism and text alignment compared to other methods.

**Strengths:**

1. The paper presents a novel perspective on zero-shot HOI generation. By leveraging Large Language Models (LLMs) to distill structured affordance knowledge , it effectively bypasses the need for limited 4D training data and achieves strong generalization.
2. The proposed Part Affordance Guidance (PAG) enables finer-grained control over the synthesis process. This explicit, part-level reasoning ensures more realistic interactions and accurate contact dynamics between specific human body parts and object parts.
3. The pipeline's effectiveness is proven through extensive experiments, where it generates realistic interactions and shows superior text alignment. Furthermore, the framework demonstrates excellent flexibility by successfully generalizing to complex multi-person and multi-object scenarios.

**Weaknesses:**

1. The pipeline is heavily dependent on a complex cascade of pre-trained models (e.g., LLMs, T2V, segmentation). The framework's quality is bottlenecked by these components, and a failure in an intermediate step, such as poor video generation or segmentation, can cause the entire result to fail.
2. The physical realism of the final motion is not guaranteed. The system is optimized to fit a *generated reference video*, which itself may lack physical plausibility. Furthermore, the decoupled optimization—estimating human motion first, then fitting the object—can lead to physical inconsistencies.
3. The generation process is highly complex and inefficient. The multi-stage pipeline requires significant computation time and multiple optimization runs for a single sample, making it unsuitable for rapid, large-scale generation.
4. The system's robustness is a concern as it relies on a trial-and-error video generation step. The paper does not sufficiently address how to handle persistent generation failures or manage the unpredictable, high computational cost this introduces.
5. Typo: L160, an → a LLM based

**Questions:**

1. The work's technical contribution is limited as it relies heavily on a complex cascade of existing large models. Future research should focus on developing a more unified, end-to-end architecture that internalizes affordance reasoning rather than depending on external, black-box components.
2. The current pipeline is overly complex, ad-hoc, and impractical. Its multi-stage, trial-and-error nature makes the generation time unpredictable and computationally expensive. A significantly streamlined and more robust framework is needed for practical application.

---

### Official Review · Reviewer_o7Ft · 2025-11-03

**Soundness:** 3
**Presentation:** 2
**Contribution:** 3
**Rating:** 4
**Confidence:** 4

**Summary:**

This paper presents an approach for generating human-object interactions (HOIs) in 3D. Part-level affordance (contact) are estimated over both object surfaces and human bodies using an LLM, which are then used as guidance for object part segmentation and 2D HOI video generation. Finally, 3D HOIs are obtained by optimizing an objective function so that the generated 3D HOIs are consistent with the 2D observations.

**Strengths:**

1. Learning object affordance is a challenging task. The idea of using LLMs to construct part affordance graphs (PAGs) is an appealing idea to leverage the semantic prior of a foundation model. The structured representation in a PAG is interpretable, compositional, and may be helpful for other applications (e.g., in robotics) as well.

2. The proposed zero-shot HOI generation framework is appealing as the motion capture data of this task is inherently limited. Using the priors of a video diffusion model can mitigate this issue.

**Weaknesses:**

1. It is not clear how the generated 4D HOI videos may follow the prompts of objects and the affordance.
   - The structured representations of PAGs are converted into textual prompts, which are fed into a image and video diffusion model. However, it is possible that the generated motion may not follow the desired interactions between human and objects.

   - It lacks details on how to condition the generated videos on the given 3D instances.

2. The generated PAGs may lack diversity to capture the interactions between humans and objects. For instance, there are many ways to push a chair, like on the top or in the middle. More clarifications are needed here.

3. InterDreamer (Xu et al., 2024) also tackles HOI generation in a zero-shot manner using foundation models. Comparisons should be reported against this important baseline.

4. More evaluation metrics tailored for 3D HOI generation should be reported in this paper, especially on the BEHAVE dataset. For instance, FID and R-precision.

5. No visual results of generated videos on the BEHAVE dataset are provided, making it hard to gauge the effectiveness of the proposed approach in the "wild" setting.

**Questions:**

1. For video depth estimation, how to ensure the consistent scales across different video frames?

2. How to ensure the LLM's output for PAG forms a graph?

3. What are the failure patterns of the proposed approach? What is the bottleneck?

---

### Meta-Review · Area_Chair_FLwE · 2026-01-06

**Summary:**

This paper proposes HOI-PAGE, a zero-shot approach for synthesizing 4D human-object interactions (HOIs) from text prompts and 3D objects. The core innovation is Part Affordance Graphs (PAGs), structured representations distilled from LLMs that encode fine-grained contact relations between human body parts and object parts. PAGs guide a three-stage pipeline: 3D object part segmentation, reference HOI video generation, and 4D motion optimization with part-level contact constraints. Experiments show HOI-PAGE outperforms baselines in realism and text alignment, and generalizes to multi-person/multi-object scenarios.

Reviewers recognized the paper’s novel part-level affordance reasoning, zero-shot capability, and flexibility for complex interactions. Key concerns centered on technical limitations of the cascaded pipeline, novelty relative to prior work, evaluation completeness, and motion granularity. Due to the absence of rebuttal, none of the concerns were addressed and the paper was not considered ready to be accepted at the moment.

**Reviewer Concerns:**

Key Concerns Raised:
- Pipeline dependency and inefficiency (fjwo): Over-reliance on cascaded pre-trained models (LLMs, T2V, segmentation) introduces fragility; multi-stage optimization is computationally expensive and impractical for large-scale use.
- Novelty and baseline coverage (Hz1Q, o7Ft): Limited novelty in part-level affordance; missing comparisons with key zero-shot HOI baselines (e.g., InterDreamer, ZeroHSI).
- Evaluation gaps (o7Ft, LToL): Lack of BEHAVE dataset visual results, insufficient metrics (e.g., FID, R-precision), and no validation of PAG accuracy/object part segmentation alignment.
- Motion quality (LToL, Hz1Q): Poor fine-grained contacts (e.g., hand-object grasps), unresolved penetration issues, and unstable object motions.
- Technical ambiguities (o7Ft): Unclear how 3D object instances condition video generation, inconsistent depth scales across video frames, and PAG structural validity.

Addressed in Rebuttal:
- None (no author rebuttal provided in the materials).

Outstanding:
- All key concerns remain unaddressed due to the absence of a rebuttal. Critical gaps in baseline comparisons, evaluation metrics, technical clarifications, and motion quality improvements are unresolved.

**Reviewer Scores:**

Due to the absence of rebuttal, all scores are likely to remain.

o7Ft: 4 → Remains 4

fjwo: 4 → Remains 4

Hz1Q: 6 → Remains 6

LToL: 4 → Remains 4

---

### Decision · Program_Chairs · 2026-01-26

Reject